# Technical Note: GRACE-compatible filtering of water storage data sets via spatial autocorrelation analysis

Ehsan Sharifi[1,2], Julian Haas[1], Eva Boergens[1], Henryk Dobslaw[1] and Andreas Güntner[1, 3]

[1]GFZ Helmholtz Centre for Geosciences, Potsdam, 14473, Germany
5  [2]Institute of Meteorology and Climate Research-Troposphere Research (IMK-TRO), Karlsruhe Institute of Technology (KIT), Karlsruhe, Germany
[3]Institute of Environmental Science and Geography, University of Potsdam, Potsdam, 14476, Germany

*Correspondence to*: Ehsan Sharifi (ehsan.sharifi@gfz.de; ehsan.sharifi@kit.edu)

10  **Abstract.** Groundwater storage anomaly (GWSA) can be derived on a global scale by subtracting various water storage compartments (WSCs) such as soil moisture, snow, surface water bodies, and glaciers from terrestrial water storage anomaly (TWSA) variations based on the GRACE/GRACE-FO satellite missions. Due to the nature of data acquisition by GRACE and GRACE-FO, filtering is essential to minimize North-South-oriented striping errors, thus resulting in a spatially smoothed TWSA signal. Nowadays, specific anisotropic decorrelation filters, such as DDK or VDK (time-variable DDK) filters, are 15  applied. For a consistent subtraction of the individual storage compartments from GRACE-based TWSA, they need to be filtered in a similar way. This study utilized WSCs from observation-based data products (glaciers, soil moisture, and snow) and the global hydrological model LISFLOOD (surface water storage) to determine a suitable filter type over the analysis period 2002–2023. Analysis revealed that the routinely used decorrelation filter, e.g. the DDK filter, introduced striping artefacts into the smoothed data and has consequently been deemed inappropriate for filtering datasets lacking GRACE-like 20  correlated error patterns. As an alternative, an isotropic Gaussian filter was chosen for further analysis. To determine the optimal filter width, an empirical correlation function was employed. By minimizing differences between the empirical spatial correlation functions of aggregated WSCs and the spatial correlation function of GRACE-based TWSA to a minimum RMSD of 0.02, an optimal filter width of 250 km was identified. This filter width could be applied to the aggregated WSCs to achieve a spatial structure similar to GRACE-TWSA, ensuring compatibility for the subtraction of WSCs from GRACE-TWSA to 25  isolate groundwater storage.

Keywords: GRACE, G3P, Groundwater, Terrestrial Water Storage, Spatial Correlation, Gaussian Filtering, Satellite Gravimetry

## 1 Introduction

Terrestrial water storage (TWS) is an important component of the Earth's water cycle, as it closes the land water balance of 30  precipitation, evapotranspiration, runoff, and TWS change. TWS is listed as one of the Essential Climate Variables (ECVs) that critically contribute to characterize the state of the Earth's climate (World Meteorological Organization et al., 2021).

Furthermore, freshwater storage as a resource is essential for life and has a pivotal role for the well-being of human societies and ecosystems. TWS is a complex variable, composed of water storage in liquid and frozen state in several compartments, including groundwater, unsaturated zone soil moisture, surface water bodies such as rivers, lakes and man-made reservoirs, snow cover, glaciers and ice caps, and biomass. The current only way to quantify and monitor TWS variations in an integrative way over all storage compartments and with global coverage is by satellite gravimetry. Since 2002, the satellite missions Gravity Recovery And Climate Experiment (GRACE) (Tapley et al., 2004) and its follow-on mission GRACE-FO (Landerer et al., 2020) (in the following both together summarized as GRACE) observe changes of the Earth's gravity field which represent mass variations and redistributions on and below the Earth surface, with TWS variations as the dominant signal for most continental areas worldwide. GRACE-based TWS anomalies (TWSA) data have seen widespread use in regional and global-scale hydrology, climatology or water resources assessment over the last two decades (see, e.g., Chen et al., 2022; Humphrey et al., 2023; Tapley et al., 2019) for an overview).

One of the important fields of application of GRACE-based TWSA in hydrology is quantifying changes of groundwater storage (see, e.g., Frappart and Ramillien, 2018 for a review). In this respect, the particular benefit of satellite gravimetry is that by observing mass changes it is the only remote sensing method that is able to monitor groundwater and to assess changes of this resource for large aquifers, independent of the depth of the groundwater table. The challenge, however, lies in the need to isolate the groundwater storage variations out of the observed TWSA variations. This requires subtracting the variations in all other terrestrial water storage compartments (WSCs) derived from other observation or model data from the GRACE-based TWSA, following a water budget approach. For this data combination, i.e., the subtraction process, it is imperative that the different WSC data sets and GRACE-based TWSA are compatible with respect to their spatial resolution. Several studies have addressed this challenge by applying spatial filtering or aggregation procedures to WSC components before subtracting them from GRACE-TWSA data. For instance, Werth et al., (2009) emphasized the importance of smoothing model-based soil moisture, snow and surface water signals to make them consistent with GRACE-scale observations. Similarly, Döll et al., (2014) applied filtering techniques to WaterGAP Global Hydrology Model (WGHM) outputs before subtracting them from GRACE-TWSA to estimate groundwater storage changes at the global scale. More recently, Ferreira et al., (2024) estimated groundwater recharge across Africa by applying spherical harmonic analysis and synthesis to ensure consistency between GRACE-derived TWSA and model-based WSCs.

However, the measurement principles of GRACE, measurement errors, and noise from various sources allow for recovering large-scale patterns of major TWSA only. At smaller spatial scales, the GRACE data are increasingly dominated by noise, including non-physical north–south-oriented stripes (see (Humphrey et al., 2023) for a review of characteristics and errors of GRACE data). Various approaches have been proposed to reduce these errors and striping patterns in order to improve the resulting TWSA for hydrological applications. In particular, for the widely used solution strategy of representing GRACE-based time-variable mass anomalies at the global scale by spherical harmonics (Wahr et al., 1998), postprocessing by applying

spatial filters to the maps of mass anomalies has been suggested. For example, the simple isotropic Gaussian filtering (Jekeli, 1981) or more advanced techniques that consider the anisotropy of the striping patterns, such as the DDK filter (Kusche, 2007) or VDK filter (Horvath et al., 2018) are applied. Both filters employ regularization based on the normal equation matrix; however, while the DDK filter uses a fixed mean normal equation matrix for all months, the VDK filter utilizes the normal

70    equations associated with each particular monthly solution. As a drawback of removing noise from the GRACE data by filtering, the already rather coarse TWSA data are further smoothed in the spatial domain and besides noise also parts of the signal of interest are removed or shifted in space. The GRACE-based TWSA data thus exhibit a high degree of spatial autocorrelation.

75    As a consequence, while global WSC data sets of, e.g., soil moisture or snow water equivalent, are often available with 0.5° or even higher spatial resolution, TWSA variations can be recovered from GRACE observations with a spatial resolution of about 300 km only. Small-scale water storage patterns that may be present in the WSC data are smoothed out in the GRACE-based TWSA (Fig. 1). Thus, for a data combination or comparison process of TWSA from GRACE with other WSC data, such as for the subtraction to groundwater storage mentioned above, it is obvious that the data sets cannot be combined as they are,

80    but adequate filtering has to be applied to the WSC data to make them compatible with GRACE data. While applying filters to simulated and observational WSC data has frequently been done in studies that also use GRACE data (e.g., Werth et al., 2009; Tian et al., 2017; Swenson, 2010), there is no general assessment or recommendation yet on which spatial filtering approach for WSC data sets is most appropriate to make them consistent with GRACE-based TWSA data.

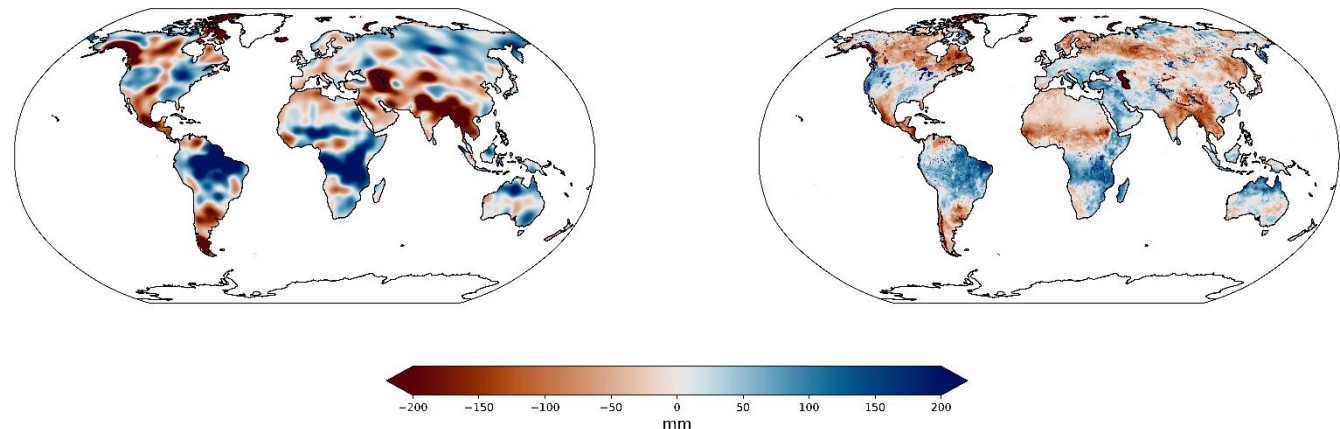

85

**Figure 1.** Terrestrial water storage anomalies (TWSA) for the example of May 2023. Left figure: TWSA based on postprocessed Level-3 data from GRACE with VDK filtering, GravIS Version 5. Right figure: Storage anomaly as the sum of the observation- and model-based WSC data sets used in this study (root-zone soil moisture, snow water equivalent, surface water storage, glacier mass change), aggregated

90    to a common 0.5° resolution, see Chapter 2 for details in the data sets.

Thus, in this study, we present a framework to identify an adequate spatial filtering approach for global WSC data sets to make them consistent with TWSA data that have been derived from GRACE Level-2 data based on spherical harmonic solutions by applying Level-3 post-processing that includes the reduction of noise and stripes. We expect that the 'optimal' filtering approach for WSC data sets varies with the type of (post-)processing of GRACE data, i.e., with their degree of smoothing, as well as with the WSC under consideration. Different WSC data sets exhibit different native patterns of storage variations that are characterized by different spatial autocorrelation functions (Güntner et al., 2007) as an expression of their smoothness. The basic concept of the framework we present here is to adjust the strength of the spatial filter (i.e., the degree to which the filter spatially smooths the WSC data) in a way that the resulting spatial autocorrelation of the WSC data sets after filtering matches in an optimal way the observed autocorrelation of the GRACE-based TWSA data. In the present study, we develop and demonstrate this framework for the example of the Global Gravity-based Groundwater Product (G3P) (Güntner et al., 2024) that uses the aforementioned subtraction approach to isolate groundwater storage variations at the global scale from GRACE-based TWSA by subtracting the storage variations of root-zone soil moisture, snow, glaciers and in surface water bodies. Nevertheless, the framework shall be transferable to other studies that aim at a consistent combination of TWSA from GRACE with WSC data sets from different sources. In addition, we perform analyses of spatial correlation characteristics of different WSCs at the global scale that may contribute to a deeper understanding of global water storage dynamics in further analyses, making this study relevant to a wider audience in hydrology and Earth system sciences.

## 2 Data

The following sections briefly introduce the global water storage data sets that are used in this study. All data sets are based on open-access data sources, partly adapted in terms of, e.g., spatial and temporal gap filling within the G3P project (Güntner et al., 2024) to allow for calculating a global data set of groundwater storage anomalies (GWSA) by the subtraction approach following the basic water budget Eq. 1:

$$GWSA = TWSA - SWE - RZSM - GM - SWS \tag{1}$$

with the water storage compartments (WSCs) root-zone soil moisture (RZSM), snow water equivalent (SWE), glacier mass (GM) and surface water storage in rivers, lakes and reservoirs (SWS). Please note that all WSCs are not expressed by their absolute storage but as time series of monthly storage anomalies relative to a long-term mean (2002-04 to 2020-12), consistent with TWSA.

### 2.1 GRACE-based Terrestrial Water Storage Anomaly (TWSA)

In this study, the combination product of seven Level-2 spherical harmonic solutions of time-variable global gravity fields from GRACE and GRACE-FO, provided by the International Combination Service for Time-variable Gravity Fields (COST-

G) (Jäggi et al., 2020; Meyer et al., 2023), is used. For the combination process, weights for the seven solutions are determined through variance components estimation on the solution level (Meyer et al., 2019). After the combination, gravity signals originating from solid Earth dynamics, i.e., glacial isostatic adjustment and the co- and post-seismic deformations from megathrust earthquakes (magnitude > 8.8) have been removed. The data set is filtered with a combination of VDK5 (deterministic part of the signal) and VDK3 (interannual and noise part of the signal). More details on the processing of the TWSA data set can be found in Dahle et al., (2025). The COST-G TWSA Level-3 product is available from April 2002 to the near present, with a nominal spatial resolution of 0.5° in units of equivalent water height at monthly time-scale.

For comparison with the COST-G-based TWSA data set described above, this study considered three other TWSA products from different processing centers, including GFZ, ITSG, and ITSG that was further processed by the University of Bonn (UB-ITSG). The GFZ data set is processed in the same processing line as the COST-G data set (Boergens et al., 2019). The ITSG data set is processed in a manner as similar as possible to the COST-G data set, with the exception of the applied filters, specifically DDK. UB-ITSG is filtered with DDK3 only and no earthquake correction is applied.

## 2.2 Snow Water Equivalent (SWE)

The Copernicus Global Land service (CGLS) Snow Water Equivalent (SWE) daily data and 0.05° spatial resolution is used. This dataset integrates information from satellite-based microwave radiometer and optical spectrometer observations with snow depth measurements at ground-based weather stations (Luojus et al., 2020; Luojus et al., 2021). Moreover, the product considers the presence of sub-grid lake ice and an improved forest transmissivity model, along with spatio-temporal snow density fields (Takala et al., 2011). For the data set used here, a merged SWE product from satellite observations and ERA5-land model (Muñoz-Sabater et al., 2021) has been developed during the G3P project. Gaps in the satellite remote sensed product are hereby filled with the most skillful SWE product from models. The skillfulness of the selected model data has been tested by comparing all datasets to in-situ ground-based snow measurements.

## 2.3 Root Zone Soil Moisture (RZSM)

The Root Zone Soil Moisture product data set used in this study is produced on behalf of the Copernicus Climate Change Service (C3S) and provides a spatio-temporally gap-filled RZSM product with daily estimates on global scale and 0.25° spatial resolution. This satellite-based RZSM product is derived from the combination of active and passive products, created by using scatterometer and radiometer soil moisture products. The product represents the unsaturated zone soil moisture and has been developed based on the methodology initially proposed by (Wagner et al., 1999). The product provides volumetric soil moisture for the top 2 m by the means of the Soil Water Index (SWI) for eight characteristic T-values (1, 5, 10, 15, 20, 40, 60, 100) from the C3S Soil Moisture CDR and is validated against in-situ soil moisture measurements from the International Soil Moisture Network (ISMN) (Pasik et al., 2023; Pasik et al., 2021; Dorigo et al., 2017)

**2.4 Glacier Mass (GM)**

The data set used is a gridded version of the C3S glacier mass product with yearly and 0.5° spatio-temporal resolution. The product is derived from the annual glacier mass change results for all of the 19 first-order glacier regions (Zemp et al., 2019; World Glacier Monitoring Service, 2018), and data are available from 1961 until near present. Annual glacier mass change has been linearly interpolated to monthly data to be consistent with other WSCs.

**2.5 Surface Water Storage (SWS)**

In the absence of global observation-based products of SWS, we use a model-based data set derived from the global hydrological model LISFLOOD (van der Knijff et al., 2010). LISFLOOD underpins the Global Flood Awareness System (GloFAS) of the Copernicus Emergency Management Service and is currently operated in version 4.0 following a global re-calibration (Choulga et al., 2023). GloFAS provides output at a 0.05° spatial and daily temporal resolution. For our SWS product, we extract model output and state variables from the modules for rivers, lakes, and man-made reservoirs and compile them into a harmonized, single-file global SWS product at 0.5° and monthly resolution.

**3 Methods**

**3.1 Pre-Processing**

The same land-ocean mask was applied to all data sets and they were made consistent in terms of units (equivalent water height in mm). Absolute storage values of each WSC were transformed into anomalies, representing deviations from the time-series mean at each grid location, in line with the GRACE TWSA data. The reference period used to calculate the mean values for anomaly generation was April 2002 to December 2020, which is the standard baseline in our data sets. In contrast, the subsequent analyses of spatial autocorrelation and filtering were carried out for the full available period, April 2002 to September 2023. This distinction ensures consistency with GRACE conventions for anomaly definition while making full use of the extended observational record for analysis. After that, all WSC data sets were harmonized to the 0.5° resolution of the TWSA data by bilinear interpolation. Bilinear interpolation ensures coverage of coastal grid cells, while conservative remapping would have led to the loss of these pixels. Similar bilinear resampling approaches have also been adopted in previous GRACE–hydrology studies (e.g., Ali et al., 2022). Furthermore, the mean seasonality (climatology) and the long-term trend were removed from all data sets and the autocorrelation analysis (see Chapter 3.3) was carried out on de-trended and de-seasonalized data. The purpose of this step is to isolate the correlation patterns that originate from GRACE-like data acquisition and processing, in particular the spatial smoothing effects, rather than from the hydro-meteorological variability itself. Seasonal or long-term signals which often have long-range spatial correlations throughout climate zones, for instance, would obscure the spatial dependencies introduced by GRACE processing. Physically, spatial autocorrelation expresses how strongly water storage anomalies at nearby locations vary with distance. High autocorrelation over long distances reflects

spatially coherent hydrological variations or strong spatial smoothing of data by filtering, while short correlation lengths indicate more localized variability or a lower degree of spatial smoothing by data processing.

This step was taken to suppress long-distance correlations that are due to the same seasonal dynamics within, e.g., the same climate zone, not of interest for the spatial autocorrelation analysis in this study (see Chapter 3.3). Finally, a combined WSC data set (called 4WSC in the following) was computed as the sum of RZSM, SWS, SWE and GM glacier, i.e., including all WSCs that are required for the subtraction process from TWSA to groundwater storage anomalies.

## 3.2 Filtering approaches

From the various filtering methods used in GRACE post-processing to remove noise and stripes from TWSA data, we selected two often used methods for filtering the WSC data sets in our analyses:

- Isotropic Gaussian filter (Jekeli, 1981): In this approach, the spatial smoothing of anomaly values is performed by a Gaussian function where the weight assigned to each point depends on the spherical distance between the kernel center and the target grid point (Fig. 2). The Gaussian filter's smoothing radius (or filter width) is defined as the distance from the center at which the Gaussian function value drops to half of its maximum (Fig. 2 right). In this study, filter widths ranging from 50 km to 600 km, in 50 km increments, were tested for smoothing the WSC data sets.

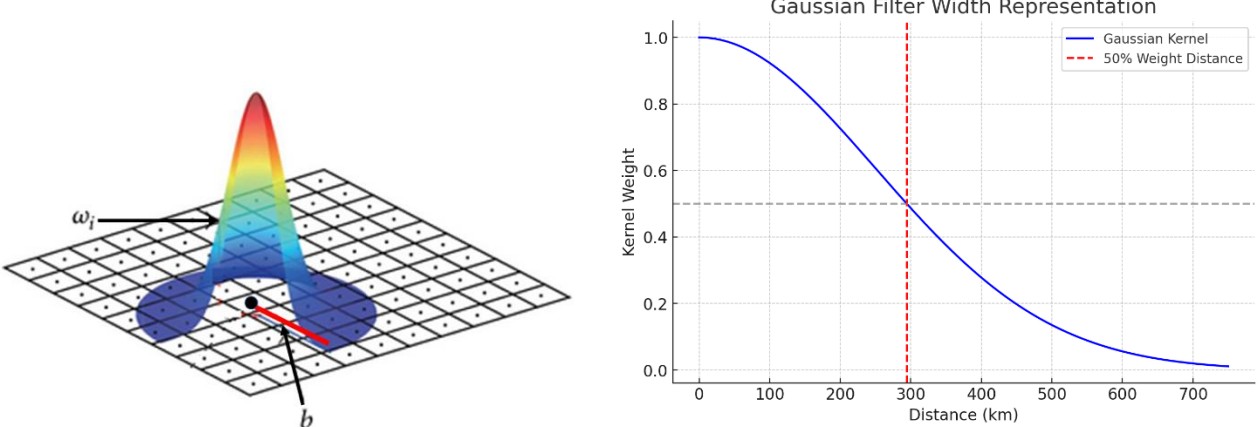

**Figure 2**. Left: Schematic representation of the Gaussian kernel function (Zhan et al., 2018) where $\omega_i$ is weight of the $i$th observation point and $b$ is the distance between the kernel center point and surrounding pixels. Right: Illustration of the Gaussian filter width, i.e., the distance where the kernel weight is at 50% (red dashed line); the blue curve represents the Gaussian kernel function, here for a Gaussian filter with 300 km filter width.

- Anisotropic decorrelation filter, also known as DDK filter (Kusche, 2007). The DDK filter is specifically designed to mitigate the non-physical north-south oriented stripes in the GRACE products by considering an anisotropic spatially correlated error structure. The DDK filter has been selected because it closely corresponds to the VDK filter

used for the GRACE TWSA set in this study. The VDK filter is unavailable for filtering the WSC data because they do not have an associated normal equation matrix that characterizes the spatial dependencies. The close correspondence between VDK and DDK filter has been exemplified by Dahle et al., (2025, Figure 2). For the application of the DDK filter, the WSC data were first converted from the spatial domain (i.e., gridded data sets) to the frequency domain using spherical harmonic analysis. Fully normalized spherical harmonic coefficients were computed up to degree and order 90, corresponding to the maximum degree and order of the GRACE products analyzed. The DDK filter coefficients were then applied in the spectral domain and the filtered fields were subsequently transformed back into the spatial domain to obtain the filtered WSC fields.

### 3.3 Spatial autocorrelation analysis

Towards defining the optimal filter width for smoothing the WSC data sets, the empirical spatial autocorrelation functions of the WSC data are compared to the empirical autocorrelation function of the GRACE-based TWSA data. For each land grid cell of the global 0.5° grid, the distances between the grid cell centers to all neighboring grid cells are determined. The spatial correlation values are then calculated for distance bins with 100-km increments in the range from 0 to 2000 km. For each time step, i.e., for all months in the study period 2002 to 2023, an empirical global autocorrelation function over all increments is then calculated by averaging the correlation values of all land grid cells in each distance increment (Eq. 2). This yields an empirical function that describes how the spatial correlation of storage anomalies decays with distance:

$$R(d) = \frac{1}{T}\sum_{t=1}^{T}\left[\frac{1}{N_d}\sum_{(i,j),(k,l)\in B_d}\frac{\sum_{t'}\left(S_{t'}(i,j)-\overline{S(i,j)}\right)\left(S_{t'}(k,l)-\overline{S(k,l)}\right)}{\sqrt{\sum_{t'}\left(S_{t'}(i,j)-\overline{S(i,j)}\right)^2}\sqrt{\sum_{t'}\left(S_{t'}(k,l)-\overline{S(k,l)}\right)^2}}\right]$$ (2)

Where:

$R(d)$ is the empirical global autocorrelation at distance bin $d$, $T$ is the total number of time steps, $t$ is the index of time step for outer averaging, $t'$ is the index of time step used for correlation calculation between pixel pairs, $(i,j)$ represents the coordinates of the first grid point, $(k,l)$ represents the coordinates of the second grid point, $S_{t'}(i,j)$ is the detrended and deseasonalized storage anomaly at time $t$ and location $(i,j)$, $\overline{(S(i,j))}$ is the mean of the time series at location $(i,j)$ over $t'$, $B_d$ is a set of all grid point pairs whose distance lies within bin $d$, and $N_d$ is the number of valid pairs in bin $d$.

The distance $Dist((i,j)(k,l))$ is computed between grid point pairs and is used to determine whether a pair belongs to $B_d$, the set of grid point pairs within distance bin $d$.

Besides this time-variable autocorrelation function, and overall autocorrelation function was calculated for each WSC and for TWSA by averaging over all time steps. It should be noted that the autocorrelation analysis carried out here is isotropic. The autocorrelation analysis was performed on a monthly basis, restricted to those months for which GRACE(-FO) TWSA data were available. WSC datasets are spatially and temporally gap-free, while GRACE(-FO) solutions may contain missing months due to instrument outages or mission gaps. These missing months were excluded from the analysis, ensuring that only valid,

consistent time steps were used for both GRACE(-FO) and WSC autocorrelation functions. Within each valid monthly field, grid cells containing missing values (e.g., ocean or masked areas) were excluded from the respective correlation pair calculations. As a result, the reference autocorrelation function is based solely on valid GRACE(-FO) data. The large number

of valid grid-cell pairs at the global scale ensures robust statistics, such that occasional (spatial) gaps do not bias the determination of the optimal filter width. Similarly, given a total number of 223 valid months for the analysis, a robust average autocorrelation with limited influence of individual missing months or the GRACE/GRACE-FO gap can be expected. It should be mentioned that at the chosen 0.5° resolution, the calculations remain computationally efficient, requiring only a few seconds per distance bin.

Furthermore, it considers sub-seasonal storage anomalies only because the data sets have been de-trended and de-seasonalized before (see Chapter 3.1). In this way it can be expected that mostly GRACE-typical correlation characteristics that origin from its data acquisition and processing are taken into account, while long-distance correlations that may exist due to similar storage dynamics within the same climate zone as suggested by (Wahr et al., 2006; Boergens et al., 2020) are eliminated.

To give a quantitative metric of the characteristic correlation length for each autocorrelation function, we employed the empirical stretched exponential model , also known as the Weibull model (Eq. 3), to describe the decay of the autocorrelation with distance (Lukichev, 2019; Mauro and Mauro, 2018).

$$C(d) = exp\left[-\left(d/\lambda\right)^{\beta}\right] \qquad\qquad (3)$$

where $C(d)$ is the autocorrelation at distance $d$, $\lambda$ is the scale parameter (correlation length), and $\beta$ is the shape parameter (decay

rate). Several candidate functions were initially tested to represent the decay of autocorrelation with distance, including simple exponential, Gaussian, inverse distance, logarithmic, and stretched exponential (Weibull) decay functions. Because of the steep decline of the autocorrelation function from the first to the second distance bin, the simple exponential model often failed to provide an adequate fit, particularly for SWE and SWS. In contrast, the Weibull model, which introduces the additional shape parameter $\beta$, consistently achieved higher $R^2$ values and provided the most robust fit across all WSCs. Therefore, the Weibull

model was selected for estimating correlation lengths in this study.

For each dataset, we first estimated the shape parameter ($\beta$) by fitting the Weibull model to the empirical correlation function for each month. The temporal mean $\beta$ value for each dataset was then used as a fixed parameter and the model was again fitted to the correlation function of each month by adjusting the correlation length parameter ($\lambda$). Using a fixed $\beta$ assured

comparability of $\lambda$ across the months within the same dataset. The stretched exponential model provides more flexibility in representing the observed patterns in the correlation functions, as it can account for variations in the shape of the correlation decay with distance. Simpler models used for estimating the correlation length, such as the exponential decay model (Güntner et al., 2007) for instance, are not able to adequately represent the steep decay of the correlation for short distances as observed for the glacier mass anomalies and SWSA (see Chapter 4.1).

For the distribution of autocorrelation values of all land grid cells in a distance bin, an empirical 2-sigma criterion was employed to determine the significance of the correlation value: distance bins for which the lower boundary of the 2-sigma range has a correlation value of 0 or smaller are considered to have a non-significant correlation value and, thus, are discarded from the further analysis (see Fig. 3).

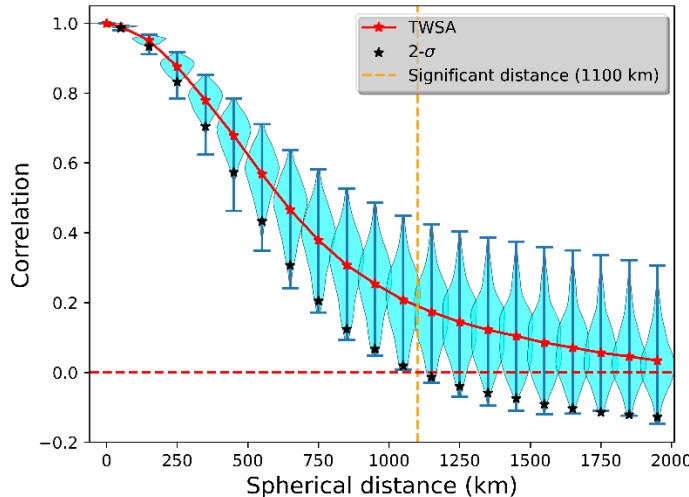


**Figure 3**: An example of an empirical spatial correlation function derived from the GRACE-based TWSA data. Violin plot of the distribution of the correlation values within each distance bin for all land grid cells. The red center point depicts the mean value for the bin, while the whiskers and the black stars represent the extreme values and the lower 2-sigma values, respectively. Distance bins with significant autocorrelation values according to the 2-sigma criterion are left to the vertical line, here at a distance of 1100 km.


As a quantitative measure of how close the empirical spatial autocorrelation functions of a WSC data set is to the empirical autocorrelation function of the GRACE-based TWSA data, a Root Mean Square Differences (RMSD) metric is calculated. The RMSD is computed from the differences between the correlation values of the WSC and TWSA for all distance bins that are significant according to the above 2-sigma criterion. The RMSD formula is given by Eq. 4:

$$RMSD = \sqrt{\frac{1}{N}\sum_{i=1}^{N}(y_i - x_i)^2}$$    (4)

Where:

$y_i$ represents the mean correlation value of the TWSA data at distance bin $i$, $x_i$ represents the mean correlation value of the WSC data at distance bin $i$, and $N$ is the number of significant distance bins.

For the WSC data, the spatial autocorrelation analysis and RMSD calculation is repeated for each filter width. The filter width

with the smallest RMSD is considered as the optimal one, as it leads to global spatial correlation characteristics of the WSC that are closest to GRACE-based TWSA.

Figure 4 summarizes the processing chain of this study to sort out the optimal GRACE-like filter method for WSC data sets.


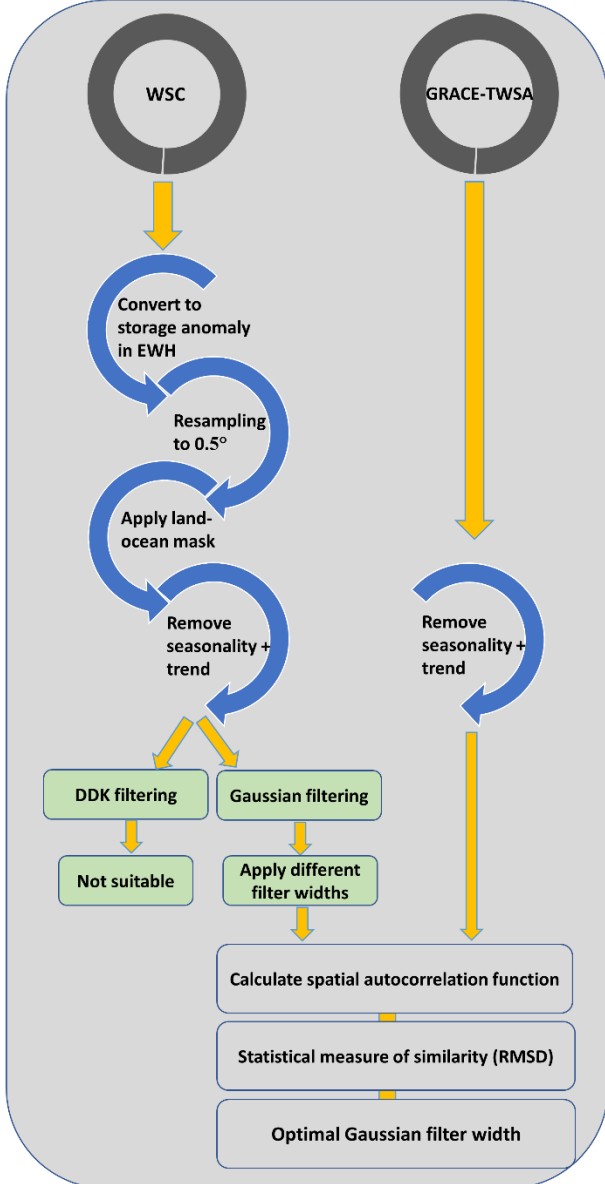

**Figure 4.** Schematic overview of the processing chain of the study towards a GRACE-like filter for WSC data.

## 4 Results and Discussion

### 4.1 Spatial autocorrelation of WSC data


The global empirical autocorrelation functions of the different WSCs and of TWSA exhibit markedly distinct characteristics (Fig. 5). The highest spatial autocorrelation among the WSCs is found for RZSM, followed by SWE, while the correlation of glacier mass anomalies and in particular of SWSA between grid cells decreases very rapidly with distance. The latter two WSCs exhibit a noticeable decline already within the first 100 km and very small correlation lengths (Table 1).

The gradual decline of correlation over several hundreds of kilometers for SWE and soil moisture indicates their relatively homogeneous spatial patterns that are governed by larger-scale hydro-climatological dynamics of precipitation, temperature, and evapotranspiration, even at the sub-seasonal time scale. In contrast, the correlation for glaciers and SWS is an expression of their high spatial variability which mainly arises from their localized nature and small spatial extent: Glaciers are typically restricted to comparatively small high-altitude regions, and SWS (e.g., rivers, lakes, reservoirs) tends to be either of a narrow

linear structure along the river network or highly discontinuous and scattered distribution in the case of lakes. These findings are similar to those of Güntner et al., (2007). The correlation length of the storage data set that was computed as the sum of the four individual WSCs (4WSC) results in an intermediate value of 306 km (Fig., 5, Table 1).

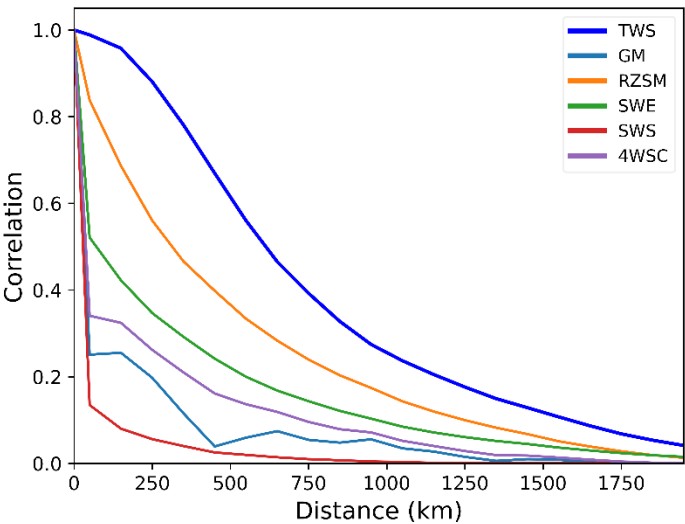

**Figure 5.** Global empirical autocorrelation functions for the different water storage compartments (WSC) and of GRACE-based TWSA (COST-G product). The curves represent temporal average autocorrelation functions for monthly data in the period 2002-2023. All data sets were de-trended and de-seasonalized before the correlation analysis. WSC data sets are unfiltered.

It should be noted that the correlation lengths given in Table 1 are obtained by fitting the Weibull model to the correlation

function (see chapter 3.4). From this, correlation lengths smaller than the resolution of the input data of 0.5° may be obtained for rapidly decaying correlation functions, i.e., for glacier mass anomalies and SWS. While these are numerical results of model fitting, the values can be considered to be of a realistic order of magnitude, given the small-scale nature of the respective

storage compartments as discussed above. Nevertheless, further studies may revisit the validity of these correlation lengths by using data sets with higher spatial resolution. It should also be noted that our analysis is addressing spatial scales of 0.5° and larger. Sub-grid scale heterogeneity, such as topographically driven water storage variations at the hillslope scale, while hydrologically important, is not captured at this resolution and is not subject of this study which covers the smoothing effects of GRACE-like data processing at large spatial scales.

**Table 1:** The correlation length (parameter $\lambda$ in Equation 2) of the water storage data sets.

| WSCs | RZSM | SWE | GM | SWS | 4WSC | TWSA |
|------|------|-----|-----|-----|------|------|
| Correlation length | 481 km | 205 km | 32 km | 5 km | 306 km | 636 km |

The spatial autocorrelation of GRACE-based TWSA with a correlation length of 636 km is considerably larger than any of the individual WSCs or the combined 4WSC (Table 1, Fig. 5). Although GRACE-based TWSA also encompasses groundwater which is not part of 4WSC, this result illustrates the degree of smoothing – or the lack of spatial detail - that the GRACE TWSA data exhibit due to their acquisition and (post-)processing methods. This stresses the need to make WSC data sets consistent with TWSA data in terms of spatial smoothing before combining them in a quantitative way, such as for the subtraction approach for groundwater storage variations.

As a first-order evaluation of temporal variations of the spatial autocorrelation of water storage anomalies, the mean monthly autocorrelation lengths were analyzed. The results clearly indicate that the deseasonalized data sets that have also been used for the analyses described above exhibit hardly any variations of the correlation length over the year (dashed lines in Fig. 6). For the storage data sets with the full signal (including seasonality), seasonal variations of the correlation lengths are apparent, with larger correlation in the spring and autumn seasons of the northern hemisphere (solid lines in Fig. 6). These seasonal patterns are amplified in the filtered 4WSC data and in GRACE-based TWS. While a detailed analysis of the reasons for these seasonally varying correlation structures is beyond the scope of this technical note, it can be hypothesized that it is due to higher spatial variability of water storage on large scales in the winter period, in particular by the snow cover, as well as in the summer period, in particular by more heterogeneous soil moisture patterns under overall drier conditions. Such process-based hydro-climatological effects on the correlation structure are largely reduced after de-seasonalizing the data sets, so that the spatial autocorrelation characteristics of GRACE-like data alone are carved out, as of primary relevance for the purpose of this study. The results show that taking the correlation length to be constant in time is a reasonable assumption.

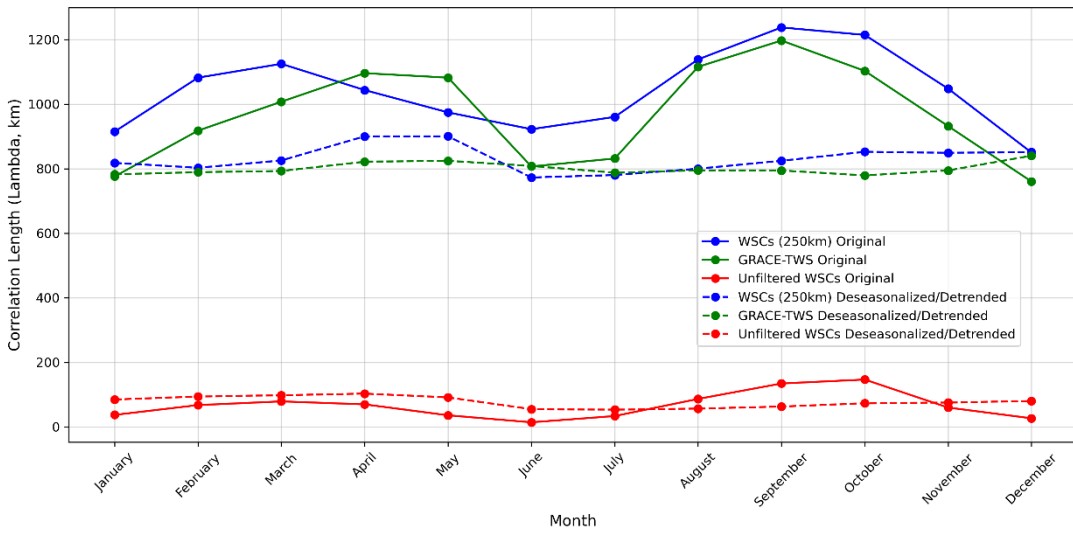


**Figure 6.** Mean monthly spatial autocorrelation lengths (based on the Weibull model) of 4WSC (unfiltered in red, 250 km Gaussian filtered in blue) and GRACE-based TWSA (in green) for the original (solid lines) and the de-seasonalized/de-trended data (dashed lines).

## 4.2 Spatial autocorrelation of TWSA from different GRACE data sets


We assessed to which extent the empirical spatial autocorrelation function of TWSA varies among different GRACE data sets that are based on spherical harmonic solutions. Among the four different data sets analyzed here, the autocorrelation functions are very similar to each other, in particular for significant correlation values at distances smaller than about 1000 km (Fig. 7). These results indicate that a correlation function based on COST-G data can readily be used as a reference in this study. The

results for optimal GRACE-consistent filtering in terms of spatial autocorrelation can be expected to vary marginally only when using another spherical harmonic solution as a basis for the TWSA data.

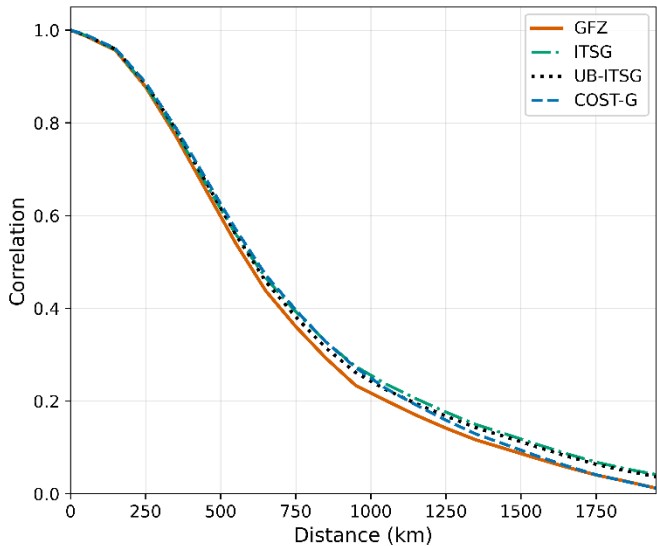

**Figure 7:** Global spatial autocorrelation functions of four different Level-3 GRACE TWSA data sets, calculated as the temporal average over the period 2002-2023.

### 4.3 Suitable filter type

Towards identifying a suitable GRACE-consistent filtering approach for WSC data sets, we first tested the anisotropic DDK filter and the isotropic Gaussian filter. Applying the DDK filter to the WSC data sets introduced striping artifacts in the filtered data (Fig. 8a), similar to those spurious features in GRACE Level-2 data that the DDK filter aims to reduce. Further, the transformation into the spherical harmonic domain and back introduced Gibbs effects, e.g., visible along the Alaska coast, around Iceland, or in the Andes. Similar results have been obtained for other WSCs. Applying the Gaussian filter resulted in an isotropic smoothing of the input data with reasonable spatial patterns (Fig. 8b). This outcome leads us to the conclusion that the DDK filter is not appropriate for filtering water storage datasets that lack GRACE-like correlated error patterns. As a consequence, the DDK filter was discarded in the following analyses and the focus was laid on optimizing the filter strength of the Gaussian filter.

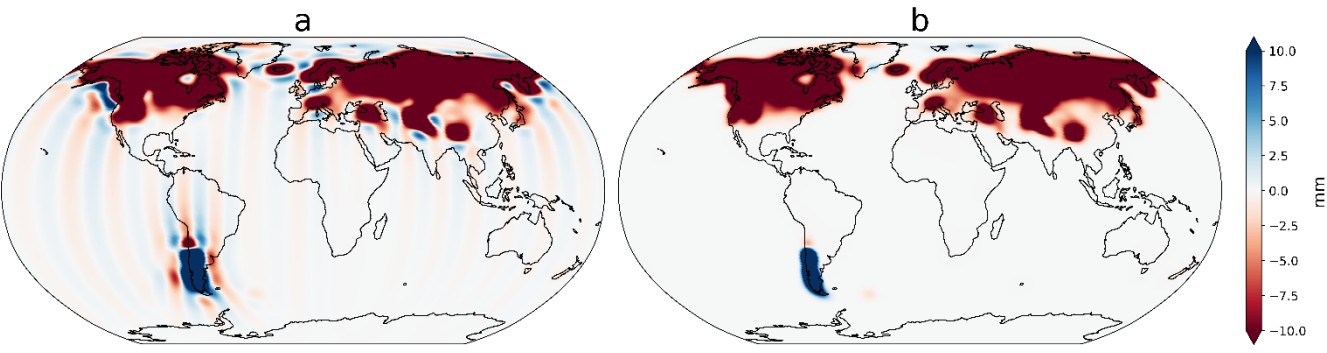

**Figure 8.** SWE anomaly of month 2002-04 (in mm water equivalent) filtered with (a) DDK3, and (b) Gaussian filter (250 km filter width)


## 4.4 Optimal Gaussian filter width

For assessing the most suitable filter width of the Gaussian filter, we looked for the minimum RMSD value when comparing the autocorrelation functions of differently filtered 4WSC to the TWSA autocorrelation function (see Chapter 3.3). An optimal filter width of 250 km was identified (Fig. 9, Table 2). With this filter width, the decay of spatial autocorrelation of 4WSC is very similar to GRACE-based TWSA, in particular for the distance range up to 1100 km where the limit of significant correlation values has been found for this analysis (see Chapter 3.3). It should be noted that the RMSD values show a relatively broad minimum between 200 km and 300 km, indicating that filter widths within this range yield similarly good results. The

selection of 250 km thus reflects the lowest RMSD in this analysis and is thus an objective guiding value, but it is suggested to re-apply the methodology for other datasets or time periods in future studies, as the optimal width may vary (see also conclusions chapter)

**Table 2:** RMSD values of the autocorrelation function of Gaussian-filtered 4WSC for different filter widths compared to the autocorrelation
function of GRACE-based TWSA. Optimum filter width in bold.

| Filter width (km) | unfiltered | 50 | 100 | 150 | 200 | **250** | 300 | 350 | 400 | 450 | 500 | 550 | 600 |
|---|---|---|---|---|---|---|---|---|---|---|---|---|---|
| RMSD | 0.33 | 0.20 | 0.12 | 0.07 | 0.03 | **0.02** | 0.05 | 0.08 | 0.11 | 0.14 | 0.17 | 0.20 | 0.22 |

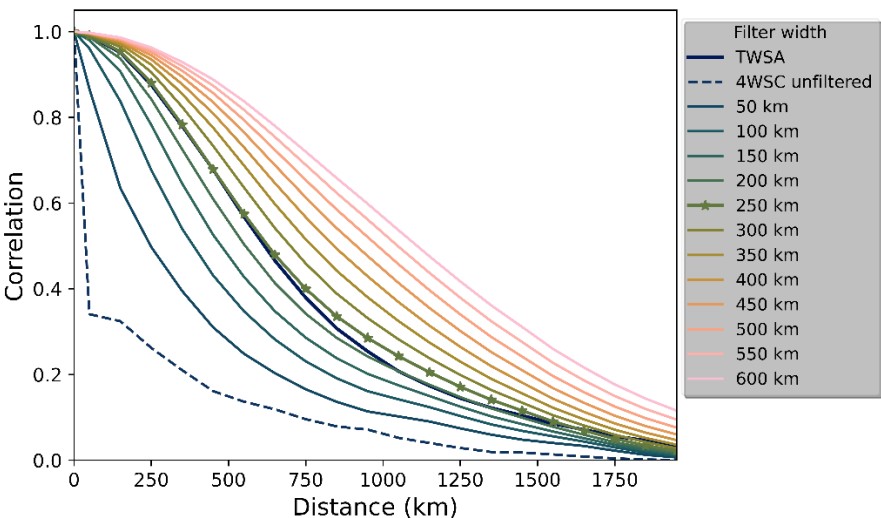

**Figure 9:** Empirical global autocorrelation functions of 4WSC after Gaussian filtering with different filter widths, compared to the GRACE-based TWSA autocorrelation function.

Given the different spatial autocorrelation patterns of the different unfiltered WSCs (see Fig. 5), different filter widths are required to make them consistent with the correlation structure of GRACE data. When applying the optimization approach as described for 4WSC to each WSC individually, the resulting optimal filter widths range from a minimum of 150 km for RZSM to a maximum of 350 km for SWS, see Table 3.

**Table 3:** Optimal filter width of all WSCs, and the RMSD values of the respective autocorrelation function when compared to the autocorrelation function of GRACE-based TWSA.

| WSCs | SWE | SWS | RZSM | GM | 4WSC |
|---|---|---|---|---|---|
| Optimal filter width | 200 km | 350 km | 150 km | 300 km | 250 km |
| RMSD | 0.03 | 0.03 | 0.03 | 0.02 | 0.02 |

In general, the results show that WSCs with smaller spatial autocorrelation in their unfiltered version require a stronger filter to make them spatially consistent with the GRACE-based TWSA data. This suggests that the degree of filtering depends on the spatial variability of each WSC. Consequently, the selection of an appropriate filter width should be tailored to the specific analysis, i.e., which WSC data sets should be compared with or reduced from GRACE-based TWSA.

### 4.5 Technical notes on computational efficiency

For the standard approach in this study, all WSC data sets were resampled to a common 0.5° resolution before applying the GRACE-like filtering approach (Chapter 3.3). However, the original resolution of some individual WSC data was higher, with a 0.25° resolution of the SWE data, for instance. We therefore tested the effect of different input resolutions on both the computational performance and the filtering results. The resulting spatial patterns and the autocorrelation functions produced nearly identical results (Fig. 10) with no discernible effect on the optimal filter width. Preprocessing steps (e.g. anomaly calculation, unit consistency, and grid harmonization) at 0.5° required 0.58 minutes and 1.7 GB memory, while the same step at 0.25° resolution took 2.4 minutes and 6.3 GB. For the correlation length calculation across all 223 months and one example distance bin (e.g. 300–400 km), processing at 0.5° required 38 seconds and 13 GB memory, whereas at 0.25° it took 998 seconds and 179 GB. This means that Processing at 0.5° resolution was approximately 26 times faster and required about 14 time less memory compared to 0.25°, while yielding identical results. The gap widens even further when considering all distance bins (0–2000 km).

Based on these results, we recommend resampling WSC data to a common lower spatial resolution (0.5°) prior to filtering, as this approach ensures identical scientific results while substantially reducing computational costs.

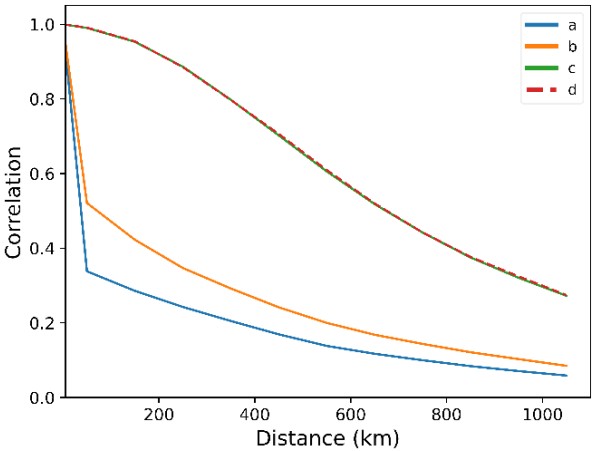

**Figure 10:** Global spatial autocorrelation functions of snow water equivalent (SWE) across different spatial resolutions and filtering sequences. The four lines represent (a) the unfiltered SWE data at 0.25° spatial resolution, (b) the unfiltered SWE data at 0.5° resolution, (c) the 0.25° SWE data filtered with a 250 km Gaussian filter, and (d) the SWE data resampled to 0.5° resolution and then filtered with a 250 km Gaussian filter.

In a second experiment, we compared the 4WSC data filtered with the Gaussian filter of 250 km width with an alternative approach in which the Gaussian filter with a 250 km width was first applied to all WSCs individually and then the filtered WSCs were combined. As shown in Fig. 11, the two resulting filtered storage patterns were practically identical. This result is not surprising, as mathematically the two approaches are expected to yield the same outcome. Given its lower computational effort, we therefore opt for the first approach that requires filter application only once on the combined 4WSC data set.

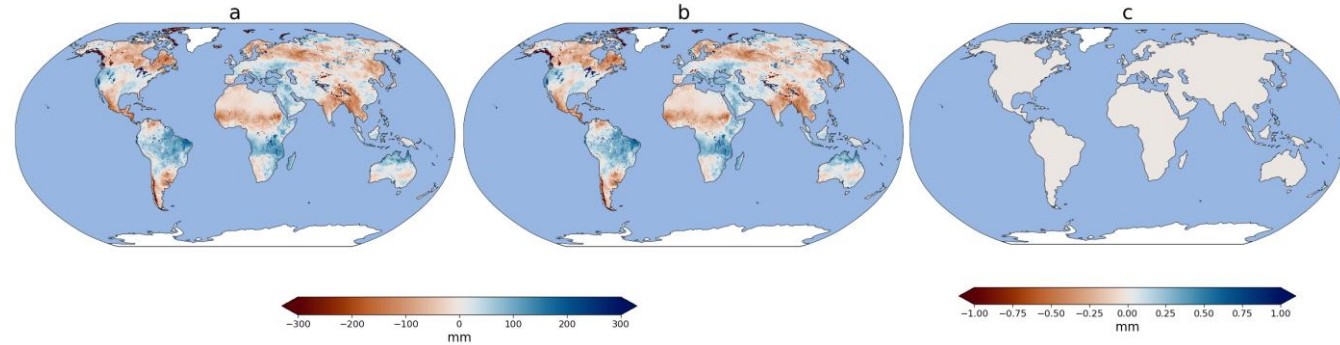


**Figure 11.** (a) 4WSC data set after 250 km Gaussian filtering, (b) Combined WSCs after individual 250 km Gaussian filtering with the original WSC resolution as input, (c) Difference between approaches (a) and (b). All plots for the example of May 2023.

**Conclusions**

With this technical note, we present a methodology to determine an optimal spatial filtering approach for global data sets of water storage variations to make them compatible in terms of their spatial resolution with coarse-resolution and spatially smoothed terrestrial water storage anomaly Level-3 data products derived from satellite gravimetry of GRACE. We find that the DDK filter commonly used for GRACE data processing to suppress anisotropic correlated noise is not suitable for other model- or observation-based Water Storage Compartments (WSCs) data sets as it introduces spurious longitudinal stripes at

the global scale. Instead, we suggest to use an isotropic Gaussian filtering kernel of which the filter width can be determined in an optimization approach by comparing the spatial autocorrelation functions of the WSC anomalies under consideration of GRACE-based TWSA.

Depending on their hydrological characteristics and spatial distribution, individual WSCs exhibit markedly different spatial autocorrelation functions and lengths. The globally and temporally averaged autocorrelation lengths of the monthly WSC

storage anomalies analyzed in this study resulted to be ordered from large to small as follows: root-zone soil moisture, snow water equivalent, glacier mass, surface water storage. The combined data set of all these four WSCs (4WSC) was found to have a spatial autocorrelation length of 306 km, in contrast to 636 km for GRACE-based TWSA. To make 4WSC spatially compatible with GRACE-based TWSA in terms of their spatial autocorrelation functions, a Gaussian filter width of 250 km was found to be the optimal value. Thus, for the primary application of interest in the background of this study, i.e., deriving

groundwater storage variations by subtracting 4WSC from GRACE-based TWSA, the 4WSC data set had to be filtered with a 250km Gaussian filter. In view of the different autocorrelation lengths of different WSCs, the selection of an appropriate filter width should be tailored to the specific application at hand. For example, for evaluating the simulated water storage of a hydrological model that does not include a glacier module with GRACE-based TWSA, glacier mass changes may need to be

removed from the TWSA data set, filtered with a GM-specific filter width. Furthermore, it should be noted that the optimal

filter methods and widths found here are obtained for TWSA Level-3 products that are derived from GRACE-solution based on a global spherical harmonic representation. Different filtering approaches can be expected to be required for TWSA based on Mascon solutions of satellite gravity observations.

In comparative analyses we furthermore found that computationally less expensive processing chains of the filtering approach can be preferred over more expensive ones as the lead to very similar results. More specifically, this includes spatial

aggregation of original WSC data sets before filtering, e.g., to a 0.5° grid, or filtering a combined data set such as 4WSC only once instead of each of its compartments separately.

The results of this study on differences in the autocorrelation characteristics between different water storage compartments and on their temporal variations illustrate the potential for further analyses towards an improved characterization and understanding of global-scale water storage dynamics both in time and space. Further studies may perform in-depth analyses

how their correlation patterns depend on physiographic or climatological factors or how and why they vary in time as a function of seasonality or of hydrological extreme events such as droughts or floods.

While this study demonstrates an approach for determining optimal GRACE-like filter widths based on spatial autocorrelation, it does neither present groundwater storage anomaly data that result through applying this approach nor evaluate the approach against independent observations. Such evaluation would require addressing additional sources of uncertainty (e.g., uncertainty

of other storage compartments, scaling issues, local hydrogeological variability, conversion of in-situ groundwater level observation to groundwater storage) and is beyond the scope of this technical note.

**Data availability**

All datasets used in this study are accessible through Güntner et al., (2024).

**Author contribution**

ES prepared the manuscript with contributions from all co-authors; AG, JH, EB, and HD reviewed and edited the manuscript; ES analyzed the data; ES, AG, JH, and EB conceptualized the study and developed and designed the methodology; ES applied formal analysis; ES produced visualizations and figures; ES, AG, JH, EB, and HD discussed the results and implications on the manuscript at all stages.

**Competing interests**

The authors declare that they have no conflict of interest.

**Acknowledgements**

This study has received funding from the European Union's Horizon 2020 research and innovation programme for G3P (Global Gravity-based Groundwater Product) under grant agreement n° 870353.

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
