# Peer review of "Technical Note: GRACE-compatible filtering of water storage data sets via spatial autocorrelation analysis"

_EGUsphere, 2025_

## Referee Comment (RC1)

**Review by Divyeshkumar Rana – 31/05/2025**

**Technical Note: GRACE-compatible filtering of water storage data sets via spatial autocorrelation analysis**

**Ehsan Sharifi, Julian Haas, Eva Börgens, Henryk Dobslaw and Andreas Güntner**

**This paper is accepted to subject a minor revision:**

Rating at 1 to 4 scale, 1 means excellent and 4 means poor score:

----------- Scientific significance ----------- SCORE: 2 (Good)

----------- Scientific quality----------- SCORE: 2 (Good)

---------- Presentation quality----------- SCORE: 2 (Good)

----------- Overall Rating ----------- SCORE: 2 (Good)

**1.1. Recommendation**

Minor Revision

**1.2. Overview**

This technical note explores methods to make Water Storage Compartments (WSCs) data compatible with Terrestrial Water Storage Anomaly (TWSA) data derived from GRACE satellite missions, specifically for isolating groundwater storage. Because GRACE data inherently have spatial smoothing and noise, other WSC datasets must be filtered similarly for accurate comparisons and subtractions. The study found that anisotropic decorrelation filters, like DDK, introduce artefacts in WSC data, suggesting that an isotropic Gaussian filter is more suitable. By analysing spatial autocorrelation and minimising differences between WSC and TWSA autocorrelation functions, an optimal Gaussian filter width of 250 km was identified for a combined WSC dataset to align with GRACE-based TWSA data characteristics.

This technical note is well written, and the overall quality of the manuscript is excellent. The research question is clearly defined and addressed in a scientifically sound manner. However, to enhance the scientific rigour of the work, I have a few specific comments regarding certain aspects of the study:

**1.3 Minor comments**

Abstract (line 20): Please include the RMSD results corresponding to the optimal filter width, as this will help emphasise the most significant findings for the reader.

Abstract: The time period of the data is not mentioned in the abstract. Please include it to provide readers with an immediate contextual understanding of the study.

Line 155: I would have appreciated more information regarding the choice of bilinear interpolation. Kindly consider including a specific reference and a clear justification for its use, particularly in relation to the data characteristics and the goals of the analysis.

Line 194: The manuscript mentions the time period as 2002 to 2023 in one instance, while the data preprocessing section refers to 2002 to 2020. Please ensure consistency throughout the manuscript regarding the time period to avoid confusion and maintain clarity.

Line 195: I would have liked to have more information on the spatial autocorrelation method employed in the study. It is not clearly described in the text, nor is a reference provided. Please consider elaborating on the approach used—such as whether it is based on Global Moran's I or another technique—and include an appropriate citation to support the methodology.

Line 196: I would have liked to see equation numbers included throughout the manuscript, as currently the equations are only referred to in the text without numbering. Including equation numbers would improve clarity and allow for more precise referencing within the manuscript.

Line 242: For the sake of consistency, I would suggest using "RMSD" in Equation 4, as it aligns with the terminology used throughout the manuscript. This will help maintain uniformity and avoid potential confusion for the reader.

Line 250: I would have liked to see the Fig. 4 flowchart improved through a clearer colour scheme and more concise text to enhance readability and understanding.

Line 330: I would have liked to see the projection system specified for the global maps presented in Figure 8. Including this information would enhance the reproducibility and clarity of the spatial analysis.

Line 390: I would have liked to see proper basemap credits included in Figure 11. Acknowledging the source of the basemap is important for transparency and adherence to data usage guidelines.

I would recommend a thorough grammatical review of the manuscript, with particular attention to sentence structure and overall presentation. Enhancing linguistic clarity and

coherence will significantly improve the readability and professional quality of the work. A few examples of:

(i)          have been suggested change to "has been suggested" (line 56)

(ii)          to change to "with" etc.. (line 70)

I would suggest including an abbreviation table at the end of the manuscript, as numerous technical terms and abbreviations are used throughout. This will enhance clarity and assist readers in understanding the content more easily.

In conclusion, I recommend that the study be accepted, subject to the minor corrections outlined above, to enhance its scientific rigour and clarity.

---

## Author Comment (AC1)

We thank the reviewers for their constructive feedback. In the following, reviewers' comments are shown in black, and our responses are provided in blue for clarity.

General comments:

This technical note entitled "GRACE-compatible filtering of water storage data sets via spatial autocorrelation analysis" addresses an important practical question in GRACE data processing: How to filter water storage component (WSC) datasets to make them compatible with GRACE-derived terrestrial water storage anomalies (TWSA)? While the topic is relevant and the approach is systematic, the note needs improvements in validation and methodological rigor before potential publication in HESS.

- Thank you very much for your thoughtful review and for recognizing the relevance and systematic nature of our study. We greatly appreciate your constructive feedback on methodological rigor and validation. In the revised manuscript, we have clarified and strengthened the methodological descriptions, provided additional justification for key choices, and addressed each of your comments in detail. While a full validation against groundwater storage anomalies is beyond the scope of this technical note, we would like to emphasize that this aspect is being addressed in a separate manuscript currently under preparation, in which we evaluate the groundwater storage product derived from our approach. We believe that together, both studies will provide a comprehensive picture of the methodology and its application.

Major issues:

1. Lines 15-17: The claim that DDK filters "introduced striping artefacts" needs quantitative support. Show correlation coefficients or other metrics demonstrating the inappropriateness.

- Thank you for raising this point. The phenomenon of striping artifacts induced by the DDK filter is clearly demonstrated in Figure 8a of our manuscript. These artifacts are visually apparent as north–south stripes in the filtered WSC datasets, which originally do not contain GRACE-like correlated errors. This demonstrates that, when applied to non-GRACE datasets, the DDK filter can generate spurious structures rather than remove them.
  In the GRACE literature, similar striping patterns are a well-known issue. Kusche et al. (2009) and Yi and Sneeuw (2022) both discuss how the DDK filter partially mitigates north–south striping in GRACE Level-2 data, and in some cases residual stripes remain even after filtering. While these studies focus on GRACE data, they illustrate that the DDK approach is not always fully effective in suppressing stripe noise.
  Our findings extend this picture by showing, to our knowledge for the first time, that applying DDK to non-GRACE data (such as snow water equivalent) can actively introduce striping patterns, because the underlying assumption of anisotropic correlated errors is not valid for such datasets. Thus, in this case the DDK filter is inappropriate.
  Given the clear visual evidence in Figure 8a and the conceptual link to limitations discussed in the literature, we believe that our demonstration provides sufficient support for the statement that DDK filtering can introduce artifacts when applied to WSC data.

[Figure]

Figure 8. SWE anomaly of month 2002-04 (in mm water equivalent) filtered with (a) DDK3, and (b) Gaussian filter (250 km filter width)

2. Lines 167-188: The DDK filter implementation description is unclear. How exactly were the WSC data converted to spherical harmonics? What degree/order was used? Please, add such details that are crucial for reproducibility.

- Thank you for pointing this out. We agree that the description of the DDK filter implementation required additional technical detail. In our analysis, the WSC fields were first expanded into fully normalized spherical harmonics up to degree and order 90 which corresponds to the effective resolution of the GRACE products considered in this study. This step was carried out using spherical harmonic analysis based on the standard least-squares collocation approach. After applying the DDK filter coefficients in the spectral domain, the filtered spherical harmonic coefficients were truncated at the same degree/order (90) and subsequently transformed back into the spatial domain.
  We have revised the manuscript text accordingly to clarify these details and ensure reproducibility as follow:

  "For the application of the DDK filter, the WSC data were first converted from the spatial domain (i.e., gridded data sets) to the frequency domain using spherical harmonic analysis. Fully normalized spherical harmonic coefficients were computed up to degree and order 90, corresponding to the maximum degree and order of the GRACE products analyzed. The DDK filter coefficients were then applied in the spectral domain, and the filtered fields were subsequently transformed back into the spatial domain to obtain the filtered WSC fields."

3. Lines 189-210: The de-trending and de-seasonalization approach (Lines 159-161) may remove important signal characteristics. The text justifies this as eliminating "long-distance correlations...not of interest" but this seems arbitrary. Seasonal signals are fundamental to water storage dynamics, and their removal may bias the autocorrelation analysis.

- Thank you for raising this important point. We agree that seasonal signals are fundamental components of water storage dynamics. However, the purpose of our study is not on describing the hydro-meteorological variability of the WSCs themselves, but on characterizing the smoothing and spatial autocorrelation properties that are induced by GRACE-like processing.

Leaving seasonal and trend components in the data would cause the autocorrelation function to be dominated by these strong, often long-range, hydrological signals, making it difficult to isolate and interpret the correlation structures that arise specifically from GRACE-type processing characteristics. By removing seasonality and trends, we minimize these dominant hydrological effects and ensure that the analysis more clearly reflects the degree of smoothing imposed by GRACE-like filtering. To clarify this, we have revised the manuscript to better explain our motivation for de-trending and de-seasonalizing prior to autocorrelation analysis as follows:

"Furthermore, the mean seasonality (climatology) and the long-term trend were removed from all data sets and the autocorrelation analysis (see Chapter 3.3) was carried out on de-trended and de-seasonalized data. The purpose of this step is to isolate the correlation patterns that originate from GRACE-like data acquisition and processing, in particular the smoothing and decorrelation effects, rather than from the hydro-meteorological variability itself. Seasonal or long-term signals which often have long-range spatial correlations throughout climate zones, for instance would obscure the spatial dependencies introduced by GRACE processing. Physically, spatial autocorrelation expresses how strongly water storage anomalies at nearby locations resemble each other as a function of distance. High autocorrelation over long distances reflects spatially coherent hydrological variations or strong spatial smoothing of data by filtering, while short correlation lengths indicate more localized variability. In this study, we use these properties primarily to quantify GRACE-related smoothing effects, not to interpret the hydrological processes underlying the seasonal or long-term variability or a lower degree of spatial smoothing by data processing."

4.  Lines 195-210: The autocorrelation calculation (Equation 2) appears computationally expensive and may be sensitive to data gaps. Specifically, GRACE TWSA data contains missing values and temporal gaps (e.g., during the gap between GRACE and GRACE-FO missions, instrument failures, or data quality flags), yet the text does not address how these gaps affect the reference autocorrelation function calculation. Additionally, individual WSC datasets may have their own spatial or temporal gaps. The text should clarify: (1) How are missing values handled when computing correlations between grid cell pairs? (2) Are months with insufficient data coverage excluded from the analysis? (3) How sensitive is the optimal filter width determination to the presence of data gaps? (4) Given that the method aims to match WSC autocorrelation to GRACE autocorrelation, how do gaps in the GRACE reference data affect the reliability of the "target" autocorrelation function itself? Without proper gap handling procedures, the computed autocorrelation functions may be biased, potentially leading to suboptimal filter width selection. (5) What about grid cells with "insufficient" neighbors?

- Thank you for raising this point. We would like to clarify that the correlation analysis was carried out on a monthly basis, and only months with available GRACE(-FO) TWSA data were included, both for GRACE-based TWSA and for all other WSC data sets. Thus, temporal gaps in GRACE (e.g., during the mission gap or flagged months) were automatically excluded from the analysis.

Regarding the specific questions:

1.  **Missing values**: Grid cells containing missing data were excluded from the respective correlation pair calculations for all WSC data sets.

2.  **Months with insufficient coverage**: Months without GRACE(-FO) solutions were not considered in the autocorrelation analysis, ensuring consistency between GRACE and WSC autocorrelation functions.

3. **Sensitivity of filter width to gaps**: We did not explicitly test the effect of individual missing months through a remove–restore experiment. However, our analysis covers 223 monthly time steps between 2002–2023, so the empirical autocorrelation functions are derived from a large sample size. Occasional gaps (e.g., during the GRACE–GRACE-FO transition or isolated flagged months) represent only a small fraction of the total record. Therefore, their influence on the averaged global autocorrelation functions, and consequently on the determination of the optimal filter width, is expected to be minor. In addition to that, since only valid GRACE months were included for all WSCs, the optimal filter width is based on a consistent reference and is not biased by incomplete data.

4. **Reliability of GRACE reference**: The "target" autocorrelation function is calculated only from months with valid GRACE(-FO) fields, meaning the reference is based on reliable data only. Furthermore, as mentioned in point 3. above, given an overall number of 223 valid months for the analysis, the impact of missing months can be expected to be small. In particular, the results can be expected to be robust as a missing January field in a particular year, for instance, is represented by about 20 other January fields in the entire time series for which similar spatial autocorrelation characteristics can be assumed.

5. **Grid cells with insufficient neighbors**: The binning scheme naturally excludes pairs with missing neighbors. As we average over global land cells, the large number of valid pairs ensures robustness.

Regarding computational expense: since our spatial resolution is coarse (0.5°), the pairwise autocorrelation calculation (Eq. 2) is not computationally expensive. Each distance bin for one monthly field can be computed within a few seconds, and the full autocorrelation function across all bins for one month typically takes only 1–2 minutes.

We have clarified the above issues in the revised manuscript as follows:

"The autocorrelation analysis was performed on a monthly basis, restricted to those months for which GRACE(-FO) TWSA data were available. WSC datasets are spatially and temporally gap-free, while GRACE(-FO) solutions may contain missing months due to instrument outages or mission gaps. These missing months were excluded from the analysis, ensuring that only valid, consistent time steps were used for both GRACE(-FO) and WSC autocorrelation functions. Within each valid monthly field, grid cells containing missing values (e.g., ocean or masked areas) were excluded from the respective correlation pair calculations. As a result, the reference autocorrelation function is based solely on valid GRACE(-FO) data. The large number of valid grid-cell pairs at the global scale ensures robust statistics, such that occasional (spatial) gaps do not bias the determination of the optimal filter width. Similarly, given a total number of 223 valid months for the analysis, a robust average autocorrelation with limited influence of individual missing months or the GRACE/GRACE-FO gap can be expected. It should be mentioned that at the chosen 0.5° resolution, the calculations remain computationally efficient, requiring only a few seconds per distance bin."

5. Lines 213-225: The Weibull model fitting approach needs more justification. Why was this model chosen over simpler exponential decay (I guess the difference would be the shape parameter). What are the goodness-of-fit statistics?

- Thank you for this important comment. We initially tested several models to describe the decay of the empirical autocorrelation functions, including the simple exponential, Gaussian, inverse distance,

logarithmic, double-logarithmic, and the stretched exponential decay models. However, because of the very steep decline from the first to the second distance bin, the simple exponential often produced systematic biases and poor fits, especially for SWE and SWS.

We therefore tested the stretched exponential (Weibull) model, which allows for an additional shape parameter (β). This additional flexibility provided a consistently better fit across all WSCs. To evaluate performance, we compared the models using $R^2$ as a goodness-of-fit measure. The Weibull model systematically achieved higher $R^2$ values than the simple exponential or other tested alternatives, and was therefore selected as the most appropriate model for deriving correlation lengths. Please see Figure bellow:

[Figure]

To clarify this in the manuscript, we have added a short explanation of the model selection process and a sentence highlighting why Weibull was chosen, as follow:

"Several candidate functions were initially tested to represent the decay of autocorrelation with distance, including simple exponential, Gaussian, inverse distance, logarithmic, and stretched exponential (Weibull) decay functions. Because of the steep decline of the autocorrelation function from the first to the second distance bin, the simple exponential model often failed to provide an adequate fit, particularly for SWE and SWS. In contrast, the Weibull model, which introduces the additional shape parameter β, consistently achieved higher $R^2$ values and provided the most robust fit across all WSCs. Therefore, the Weibull model was selected for estimating correlation lengths in this study."

6. A proper validation of the proposed approach is required. The study demonstrates that filtered WSCs match GRACE autocorrelation functions, but it doesn't validate whether this actually improves groundwater storage anomaly (GWSA) estimates. Does the 250 km filter width actually produce more accurate GWSA compared to other approaches? How do the filtered WSCs compare against independent observations? What is the impact on signal preservation versus noise reduction?

- - We fully agree that a validation against independent groundwater storage anomaly (GWSA) observations would be valuable. However, such validation is beyond the scope of the present study, which is intended as a methodological note focusing on the filtering of water storage compartments (WSCs) in a GRACE-like manner. Our main objective was to evaluate and demonstrate an approach for identifying optimal filter widths based on spatial autocorrelation characteristics, and not to assess the full performance of GWSA estimates themselves.
  We would like to emphasize that a direct validation against independent GWSA observations would face several challenges:

  Multiple sources of uncertainty (e.g., uncertainties of the other WSCs that are subtracted from TWSA to obtain groundwater storage anomalies, differences in local hydrogeological conditions, scaling mismatch, measurement errors in well data, conversion errors from well observations of groundwater level into groundwater storage through uncertainties of specific yield/storativity) could dominate deviations, making it difficult to attribute differences specifically to the filtering step.

  Signal preservation vs. noise reduction trade-offs are indeed important, but their evaluation requires a dedicated study with independent reference data sets, which would extend far beyond the methodological focus of this technical contribution.

  Therefore, our study does not claim to provide a full validation of the GWSA estimates. Instead, we demonstrate that, based on our data sets, the chosen 250 km filter width achieves consistency of WSCs with GRACE-based TWSA autocorrelation functions, which is a necessary step to enable subsequent, application-specific analyses of groundwater or other storage components.
  We have clarified this scope in the revised manuscript to avoid any misunderstanding, as follow:

  "While this study demonstrates an approach for determining optimal GRACE-like filter widths based on spatial autocorrelation, it does neither present groundwater storage anomaly data that result through applying this approach nor evaluate the approach against independent observations. Such evaluation would require addressing additional sources of uncertainty (e.g., uncertainty of other storage compartments, scaling issues, local hydrogeological variability, conversion of in-situ groundwater level observation to groundwater storage) and is beyond the scope of this technical note."

7.  Lines 159, 275-280: The 0.5° resolution analysis may be too coarse to capture important small-scale spatial patterns, and although the text acknowledges this limitation (Lines 275-280), it doesn't adequately address it.

- We agree that the 0.5° resolution used in our analysis does not capture fine-scale spatial patterns, such as topographically controlled water storage variations at the hillslope scale. While these patterns can be hydrologically important, they are not subject of this study which addresses the smoothing effects of GRACE-like data processing at large spatial scales. Our study is limited to the resolution of the available global WSC datasets, and our analysis is explicitly focused on scales resolvable at 0.5° and larger. We have clarified this issue in the manuscript by adding a sentence to emphasize that small-scale spatial heterogeneity below 0.5° is not addressed (and does not need to be addressed) for the objectives of this study:

  "Nevertheless, further studies may revisit the validity of these correlation lengths by using data sets with higher spatial resolution. It should also be noted that our analysis is inherently addressing spatial scales of 0.5° and larger. Sub-grid scale heterogeneity, such as topographically driven water storage variations at the hillslope scale, while hydrologically important is not captured at this resolution and is not subject of this study which covers the smoothing effects of GRACE-like data processing at large spatial scales."

8.  Lines 336-340: The optimal 250 km filter width is presented as definitive, but Table 2 shows the RMSD minimum is quite broad (200-300 km range), and the sensitivity analysis seems insufficient.

- Thank you for this valuable observation. We agree that the RMSD minimum in Table 2 is broad, with low values in the 200–300 km range, and not sharply peaked at 250 km. Within the scope of this technical note, which is based on a purely statistical analysis of spatial autocorrelation functions, we chose the global minimum RMSD value (0.02 at 250 km) as the "optimal" filter width. Thus, while we believe that 250 km is an objective and reproducible estimate, we acknowledge that this should not be interpreted as a universal or definitive value. As noted in the manuscript, the optimal filter width may vary depending on the dataset, time period, or application. Therefore, we suggest to apply the methodology also for other data sets and applications in future studies to identify the most suitable filter width. We modified the revised manuscript to clarify this point as follow:

  "For assessing the most suitable filter width of the Gaussian filter, we looked for the minimum RMSD value when comparing the autocorrelation functions of differently filtered 4WSC to the TWSA autocorrelation function (see Chapter 3.3). An optimal filter width of 250 km was identified (Fig. 9, Table 2). With this filter width, the decay of spatial autocorrelation of 4WSC is very similar to GRACE-based TWSA, in particular for the distance range up to 1100 km where the limit of significant correlation values has been found for this analysis (see Chapter 3.3). It

should be noted that the RMSD values show a relatively broad minimum between 200 km and 300 km, indicating that filter widths within this range yield similarly good results. The selection of 250 km thus reflects the lowest RMSD in this analysis and is thus an objective guiding value, but it is suggested to re-apply the methodology for other datasets or time periods in future studies, as the optimal width may vary (see also conclusions chapter)"

Minor issues:

1. Line 27: "TWS is a fundamental component" - this statement is too strong. Consider "TWS is an important component."

- Done.

2. Lines 40-45: The review of GRACE applications is superficial. Please provide a more comprehensive review of the studies that applied a filter to WSC components. For example, Ferreira et al. (2024, doi: 10.1016/j.ejrh.2024.102046) did this through spherical harmonic analysis and synthesis.

- Following your recommendation, we expanded this part by adding a more comprehensive overview of studies to provide a broader context of applied filtering to WSC as follow:

  "Several studies have addressed this challenge by applying spatial filtering or aggregation procedures to WSC components before subtracting them from GRACE-TWSA data. For instance, (Werth et al., (2009) emphasized the importance of smoothing model-based soil moisture, snow and surface water signals to make them consistent with GRACE-scale observations. Similarly, Döll et al., (2014) applied filtering techniques to WaterGAP Global Hydrology Model (WGHM) outputs before subtracting them from GRACE-TWSA to estimate groundwater storage changes at the global scale. More recently, Ferreira et al., (2024) estimated groundwater recharge across Africa by applying spherical harmonic analysis and synthesis to ensure consistency between GRACE-derived TWSA and model-based WSCs."

3. Lines 154-164: The harmonization procedure needs more detail. Bilinear interpolation may introduce artifacts; was this tested?

- We have clarified the interpolation choice in the revised manuscript. In our workflow, bilinear interpolation was applied to resample and harmonize the original WSC datasets (RZSM and SWE from 0.25° to 0.5°, and SWS from 0.1° to 0.5°) to the target resolution of 0.5° before applying the Gaussian filter. We tested the alternative sequence (filtering first at the native resolution and interpolating afterwards) and found that the results were virtually identical in terms of spatial autocorrelation. Since interpolation before filtering was computationally much more efficient, we adopted this order. Bilinear interpolation was also selected because it preserves coastal grid cells, which would be lost with conservative remapping, and provides a smooth resampling that avoids

artificial discontinuities. At the target resolution of 0.5°, these anomalies vary smoothly in space, making bilinear interpolation an appropriate choice. Similar bilinear resampling approaches have also been adopted in previous GRACE–hydrology studies (e.g., Ali et al., 2022).

The revised text in the manuscript now reads:

"Bilinear interpolation ensures coverage of coastal grid cells, while conservative remapping would have led to the loss some of these pixels. Similar bilinear resampling approaches have also been adopted in previous GRACE–hydrology studies (e.g., Ali et al., 2022)."

4. Line 125: Equation (2) is missing the number.

- Corrected.

5. Line 240: Equation (4) is missing the number, and consider using RMSD instead of RMSE for consistency throughout the text.

- Corrected.

6. Consider improving Figure 4.

- Thank you for this suggestion. We have revised Figure 4 using a clearer color scheme and updated shapes/formatting to make the flowchart more visually appealing and to improve readability and understanding.

[Figure]

Figure 4. Schematic overview of the processing chain of the study towards a GRACE-like filter for WSC data.

7. Lines 365-385: For a technical note format, the computational efficiency discussion is appropriate and adds practical value. However, this section could be strengthened by providing more quantitative details. The study demonstrates that preprocessing at 0.5° resolution yields identical results to filtering at higher resolution, and that filtering the combined 4WSC dataset once is equivalent to filtering each WSC individually, both valuable practical insights. To enhance this section, consider adding: (1) quantitative timing comparisons between the different approaches, (2) memory usage considerations for large datasets, (3) specific computational cost savings (e.g., "reduces processing time by X%"), and (4) clearer recommendations for different use cases (e.g., when computational resources are limited vs. when maximum precision is needed). These additions would make the computational guidance even more actionable for researchers implementing this filtering approach.

- Thank you for this valuable suggestion. We agree that quantitative details on computational efficiency strengthen this section. We therefore added specific timing and memory usage results comparing 0.25° and 0.5° resolution processing. For preprocessing (unit checks, anomaly calculation, coordinate harmonization, etc.), the 0.5° workflow required ~0.6 minutes and 1.7 GB memory, while the 0.25° workflow required ~2.4 minutes and 6.3 GB. For the correlation length calculation (all 223 months,

one distance bin of 300–400 km), the 0.5° workflow required ~38 seconds and 13 GB, whereas the 0.25° workflow required ~998 seconds and 179 GB. This means that the 0.5° workflow runs about 26 times faster and uses about 14 times less compared to the 0.25° workflow, while producing identical correlation length and optimal filter widths. We emphasize that when extending to all distance bins (0–2000 km), these differences become even more pronounced. We have updated the manuscript accordingly to highlight these computational trade-offs and provide clear recommendations for choosing the coarser resolution to ensure efficiency without compromising results:

"For the standard approach in this study, all WSC data sets were resampled to a common 0.5° resolution before applying the GRACE-like filtering approach (Chapter 3.3). However, the original resolution of some individual WSC data was higher, with a 0.25° resolution of the SWE data, for instance. We therefore tested the effect of different input resolutions on both the computational performance and the filtering results. The resulting spatial patterns and the autocorrelation functions produced nearly identical results (Fig. 10) with no discernible effect on the optimal filter width. Preprocessing steps (e.g. anomaly calculation, unit consistency, and grid harmonization) at 0.5° required 0.58 minutes and 1.7 GB memory, while the same step at 0.25° resolution took 2.4 minutes and 6.3 GB. For the correlation length calculation across all 223 months and one example distance bin (e.g. 300–400 km), processing at 0.5° required 38 seconds and 13 GB memory, whereas at 0.25° it took 998 seconds and 179 GB. This means that Processing at 0.5° resolution was approximately 26 times faster and required about 14 time less memory compared to 0.25°, while yielding identical results. The gap widens even further when considering all distance bins (0–2000 km).

Based on these results, we recommend resampling WSC data to a common lower spatial resolution (0.5°) prior to filtering, as this approach ensures identical scientific results while substantially reducing computational costs."

8.  Consider adding an interpretation of what the autocorrelation patterns mean physically

- Thank you for this suggestion. We agree that clarifying the physical meaning of spatial autocorrelation is helpful. Our study does not aim to provide a hydrological interpretation of the seasonal or long-term variations in autocorrelation functions, but rather to use them as a methodological tool for defining appropriate filter widths.
  To address this, we have added the following explanatory sentence to the manuscript:

  "Physically, spatial autocorrelation expresses how strongly water storage anomalies at nearby locations vary with distance. High autocorrelation over long distances reflects spatially coherent hydrological variations, while short correlation lengths indicate more localized variability. In this study, we use these properties primarily to quantify GRACE-related smoothing effects, not to interpret the hydrological processes underlying the seasonal or long-term variability."

  In line with our response to your major comment #3, we emphasize that our focus is not on describing hydrological seasonality or trends in the autocorrelation patterns themselves, but on quantifying the GRACE-like smoothing effect that is relevant for filter optimization.

---

## Author Comment (AC2)

**1.1. Recommendation**

Minor Revision

**1.2. Overview**

This technical note explores methods to make Water Storage Compartments (WSCs) data compatible with Terrestrial Water Storage Anomaly (TWSA) data derived from GRACE satellite missions, specifically for isolating groundwater storage. Because GRACE data inherently have spatial smoothing and noise, other WSC datasets must be filtered similarly for accurate comparisons and subtractions. The study found that anisotropic decorrelation filters, like DDK, introduce artefacts in WSC data, suggesting that an isotropic Gaussian filter is more suitable. By analysing spatial autocorrelation and minimising differences between WSC and TWSA autocorrelation functions, an optimal Gaussian filter width of 250 km was identified for a combined WSC dataset to align with GRACE-based TWSA data characteristics.

This technical note is well written, and the overall quality of the manuscript is excellent. The research question is clearly defined and addressed in a scientifically sound manner. However, to enhance the scientific rigour of the work, I have a few specific comments regarding certain aspects of the study.

- Thank you very much for your positive overall assessment of our manuscript and for your constructive comments. We carefully considered each of your suggestions and have revised the manuscript accordingly, addressing all points one by one to improve both clarity and scientific rigor.

**1.3 Minor comments**

Abstract (line 20): Please include the RMSD results corresponding to the optimal filter width, as this will help emphasise the most significant findings for the reader.

- Done.

Abstract: The time period of the data is not mentioned in the abstract. Please include it to provide readers with an immediate contextual understanding of the study.

- Done.

Line 155: I would have appreciated more information regarding the choice of bilinear interpolation. Kindly consider including a specific reference and a clear justification for its use, particularly in relation to the data characteristics and the goals of the analysis.

We have clarified the interpolation choice in the revised manuscript. In our workflow, bilinear interpolation was applied to resample and harmonize the original WSC datasets (RZSM and SWE from 0.25° to 0.5°, and SWS from 0.1° to 0.5°) to the target resolution of 0.5° before applying the Gaussian filter. We tested the alternative sequence (filtering first at the native resolution and interpolating afterwards) and found that the results were virtually identical in terms of spatial autocorrelation. Since interpolation before filtering was computationally much more efficient, we adopted this order. Bilinear interpolation was also selected because it preserves coastal grid cells, which would be lost with conservative remapping, and provides a smooth resampling that avoids artificial discontinuities. At the target resolution of 0.5°, these anomalies vary smoothly in space, making bilinear interpolation an appropriate choice. Similar bilinear resampling approaches have also been adopted in previous GRACE–hydrology studies (e.g., Ali et al., 2022).

The revised text in the manuscript now reads:

"Bilinear interpolation ensures coverage of coastal grid cells, while conservative remapping would have led to the loss some of these pixels. Similar bilinear resampling approaches have also been adopted in previous GRACE–hydrology studies (e.g., Ali et al., 2022)."

Ali, Shoaib; Wang, Qiumei; Liu, Dong; Fu, Qiang; Mafuzur Rahaman, Md.; Abrar Faiz, Muhammad; Jehanzeb Masud Cheema, Muhammad (2022): Estimation of spatio-temporal groundwater storage variations in the Lower Transboundary Indus Basin using GRACE satellite. In *Journal of Hydrology* 605 (2), p. 127315. DOI: 10.1016/j.jhydrol.2021.127315

Line 194: The manuscript mentions the time period as 2002 to 2023 in one instance, while the data preprocessing section refers to 2002 to 2020. Please ensure consistency throughout the manuscript regarding the time period to avoid confusion and maintain clarity.

- Thank you for pointing out this. We have clarified the text to distinguish between the reference period used for anomaly generation (2002-04 to 2020-12) and the analysis period (2002-04 to 2023-09). The former was selected to remain consistent with GRACE conventions for anomaly definition, while the latter reflects the full time span of available data used in our analyses. The revised manuscript now makes this distinction explicit to avoid confusion, as follow:

  "The reference period used to calculate the mean values for anomaly generation was April 2002 to December 2020, which is the standard baseline in our data sets. In contrast, the subsequent analyses of spatial autocorrelation and filtering were carried out for the full available period, April 2002 to September 2023. This distinction ensures consistency with GRACE conventions for anomaly definition while making full use of the extended observational record for analysis."

Line 195: I would have liked to have more information on the spatial autocorrelation method employed in the study. It is not clearly described in the text, nor is a reference provided. Please

consider elaborating on the approach used—such as whether it is based on Global Moran's I or another technique—and include an appropriate citation to support the methodology.

- Our analysis does not rely on a single-value global index such as Moran's I, but instead on the empirical spatial autocorrelation function, as described in previous GRACE–hydrology studies (e.g., Güntner et al., 2007; Boergens et al., 2020). The method involves calculating correlations of storage anomalies between all grid cell pairs as a function of their spatial separation distance, and then averaging the correlations into distance bins. This yields an empirical function describing how correlation decays with distance, from which the characteristic correlation length can be derived. The method is already described in the original manuscript in Chapter 3.3, specified by Equation (2). We added a sentence with the qualitative description of the autocorrelation function to this paragraph, and, following a comment by Reviewer 2, a description of the choice of the decay function.

Line 196: I would have liked to see equation numbers included throughout the manuscript, as currently the equations are only referred to in the text without numbering. Including equation numbers would improve clarity and allow for more precise referencing within the manuscript.

- Done, equation numbers have been added throughout the manuscript.

Line 242: For the sake of consistency, I would suggest using "RMSD" in Equation 4, as it aligns with the terminology used throughout the manuscript. This will help maintain uniformity and avoid potential confusion for the reader.

- Done.

Line 250: I would have liked to see the Fig. 4 flowchart improved through a clearer colour scheme and more concise text to enhance readability and understanding.

- Thank you for this suggestion. We have revised Figure 4 using a clearer color scheme and updated shapes/formatting to make the flowchart more visually appealing and to improve readability and understanding.

[Figure]

**Figure 4.** Schematic overview of the processing chain of the study towards a GRACE-like filter for WSC data.

Line 330: I would have liked to see the projection system specified for the global maps presented in Figure 8. Including this information would enhance the reproducibility and clarity of the spatial analysis.

- Thank you for this comment. For the global maps in Figure 8 we used the Robinson projection (ccrs.Robinson() from the Cartopy package for python). This is a widely applied standard projection for global-scale visualization, and for that reason we had not explicitly specified it in the manuscript.

Line 390: I would have liked to see proper basemap credits included in Figure 11. Acknowledging the source of the basemap is important for transparency and adherence to data usage guidelines.

- Thank you for pointing this out. The basemap features in Figure 11 (coastlines, land polygons, borders) are taken from the Natural Earth dataset, which is provided through the Cartopy package. While this is a standard open-source dataset and we did not explicitly acknowledge it in the manuscript, we agree that acknowledging the source improves transparency. We are pleased to confirm here that Natural Earth was the data source for the basemap.

I would recommend a thorough grammatical review of the manuscript, with particular attention to sentence structure and overall presentation. Enhancing linguistic clarity and coherence will significantly improve the readability and professional quality of the work. A few examples of:

(i)         have been suggested change to "has been suggested" (line 56)

(ii)        to change to "with" etc.. (line 70)

- Done.

I would suggest including an abbreviation table at the end of the manuscript, as numerous technical terms and abbreviations are used throughout. This will enhance clarity and assist readers in understanding the content more easily.

- Thank you very much for this helpful suggestion. We agree that clarity is essential given the large number of abbreviations used in the manuscript. According to the HESS author guidelines, dedicated abbreviation table in the end is not the standard format. Instead, abbreviations should be introduced and defined upon their first appearance in the text. We have carefully revised the manuscript to ensure that all abbreviations are clearly defined at first use, so that readers can follow the terminology consistently throughout.

In conclusion, I recommend that the study be accepted, subject to the minor corrections outlined above, to enhance its scientific rigour and clarity.

---

## Referee Report (RR1)

**Review Report – 27/10/2025**

Technical Note: GRACE-compatible filtering of water storage data sets via spatial autocorrelation analysis

Ehsan Sharifi, Julian Haas, Eva Börgens, Henryk Dobslaw and Andreas Güntner

This paper is accepted for Publication:

Rating at 1 to 4 scale, 1 means excellent and 4 means poor score:

----------- Scientific significance ----------- SCORE: 2 (Good)

----------- Scientific quality----------- SCORE: 2 (Good)

---------- Presentation quality----------- SCORE: 2 (Good)

----------- Overall Rating ----------- SCORE: 2 (Good)

The authors have carefully and comprehensively addressed all the comments raised during the review process. The manuscript has been significantly improved and now meets the standards for publication. I recommend the paper for acceptance. Congratulations to all the authors on their excellent work.